# Perceived Parenting Styles and Adjustment during Emerging Adulthood: A Cross-National Perspective

**DOI:** 10.3390/ijerph16152757

**Published:** 2019-08-02

**Authors:** Águeda Parra, Inmaculada Sánchez-Queija, María del Carmen García-Mendoza, Susana Coimbra, José Egídio Oliveira, Marta Díez

**Affiliations:** 1Department of Developmental and Educational Psychology, Universidad de Sevilla, 41013 Sevilla, Spain; 2Department of Psychology, Faculdade de Psicologia e de Ciências da Educação da Universidade do Porto, 4200-135 Porto, Portugal

**Keywords:** emerging adulthood, parenting, psychological wellbeing, psychological distress

## Abstract

The aim of the present study is to determine whether the influence of parenting style on children’s wellbeing is sustained during emerging adulthood. This is a stage in which young people, despite feeling themselves to be adults, often remain in the family home and continue to be financially dependent on their parents. Moreover, since parents’ beliefs, attitudes and behaviors are constructed and interpreted within their cultural milieu, the study also aims to explore the situation in Spain (SP) and Portugal (PT). Those two Southern Europe countries are representative of what is known as the “family welfare regime”, in which the family acts as the main provider of care and security not only during childhood, but also during emerging adulthood. Thus, the present study examines, from a cross-cultural perspective, the relationship between perceived parenting styles and psychological adjustment among a sample of 1047 emerging adults from Spain and Portugal. The results reveal that the most beneficial styles during this stage are the authoritative and permissive ones, with the authoritarian style being more closely related to psychological distress. The study highlights intercultural similarities and the positive role played by more symmetrical relationships in the adjustment of emerging adults in both countries.

## 1. Introduction

Parenting styles [1,2] probably constitute the most important conceptualization under which the effect of family socialization on children’s wellbeing has been studied. According to this conceptualization, parents can be divided into four groups, depending on the warmth and affection they show to their children and their level of control: authoritative (high warmth, high control), authoritarian (low warmth, high control), permissive (high warmth, low control) and neglectful (low warmth, low control).

Many recent studies have analyzed the effect of these four parenting styles during on child and adolescent wellbeing, finding, in general, that the most beneficial is the authoritative one. The children of authoritative parents have higher levels of self-esteem, moral development, interest, motivation and academic performance [3,4]; they also have an internal attributional style, consume alcohol and other drugs less frequently, are less likely to succumb to negative peer group pressure and have fewer behavioral problems [5,6,7,8]. At the opposite extreme, are the children of neglectful parents, who have more problems at school, a lower level of psychological competence and higher levels of psychological and behavioral dysfunction [7,9,10]. Midway, between the two, are those children brought up in permissive and authoritarian homes, who have difficulties at different levels [7,11,12,13].

Despite these results, however, right from the beginning some studies have challenged the universality of the positive effects of the authoritative style for child and adolescent wellbeing, questioning whether this style is always the best option in all cases. For example, various studies have highlighted the positive effect of the authoritarian [6,14] and permissive styles [15,16,17,18] for young people from different sociocultural contexts.

Moreover, classic studies carried out in the United States demonstrated years ago that parenting style varies in accordance with social and cultural reference groups, and that Asian-American and Hispanic families were more authoritarian and less authoritative than their European-American counterparts [19,20]. These studies also highlight the importance of taking social and cultural aspects into account when attempting to understand how parenting styles affect children’s wellbeing [21,22], since it should not be forgotten that parents’ childrearing beliefs, attitudes and behaviors are constructed and interpreted within their historical and cultural contexts [23].

It is important to remember that the vast majority of research into parenting styles has focused on families with children and adolescents, with only a few analyzing the impact of parenting style during emerging adulthood [24]. Emerging adulthood [25,26] refers to the period that transpires between reaching legal adulthood (at the age of 18 in the majority of countries) and age 29. This new stage emerged as the result of the social and economic changes that have occurred over recent decades, such as an increase in the number of years young people spend studying, more widespread access to university-level studies and an increase in youth unemployment, all of which have delayed the acquisition of typically adult roles. Thus, roles such as attaining stable employment, establishing a long-term romantic relationship or having children, which decades ago were acquired almost immediately after adolescence, are currently achieved nearer to age thirty [27]. Indeed, although in many ways they are fully mature, emerging adults tend to stay in their family home until well into their twenties, remaining fairly dependent on their parents during that time [28,29].

The fact that parents and their adult children live together under the same roof during this stage of the latter’s lives is a recent phenomenon that raises a series of new and interesting questions about the conceptualization of parenting style. One of the most important questions is if parenting style continues to affect young adults’ development during this phase in the same way as it did during childhood and adolescence, and if so, which style is most beneficial. The little research that has been carried out in this field has yet to offer definitive conclusions. Some studies argue that the authoritative style has a positive impact on wellbeing during emerging adulthood, leading to greater academic achievement [30], self-regulation [31], self-esteem and emotional wellbeing [32]. Nevertheless, other authors report finding no evidence of the benefits of the authoritative style during this stage, arguing that it is not associated in any way with either depressive symptoms or maladaptive behaviors [33,34]. Going one step further, in a recent study, McKinney [35] even claims that emerging adults reported fewer psychological symptoms when they rated parents as low on the authoritative scale, thereby precluding any possible positive effect of this style.

Reaching definitive conclusions is even more complicated because some studies claim that the effect of parenting style during emerging adulthood may vary in accordance with the sex of the child [18]. Thus, for example, some authors claim that the authoritarian and neglectful mothering styles are positively associated with daughters’ depressive symptoms, whereas authoritarian mothering is negatively associated with sons’ depressive symptoms [33]. Others have found that perceptions of having an authoritarian father is positively linked to higher levels of neuroticism among males but not among females, and perceptions of having a permissive father are linked to lower levels of neuroticism only among females [36].

Thus, just as during childhood, it seems that parents use different parenting styles for their sons and daughters during emerging adulthood [37,38]. Nevertheless, no consensus has yet been reached regarding the exact nature of these differences, and they are also likely to vary across cultures depending on the degree of adherence to traditional gender stereotypes. Thus, in a study carried out with Japanese youths, Someya et al. found that sons described more rejecting parenting styles with a low level of warmth, which is indicative of the authoritarian pattern, while daughters perceived more warmth and described their parents’ styles as being more caring, which is indicative of the authoritative pattern [39]. Research carried out in the United States, on the other hand, found that male emerging adults reported receiving more permissive and less authoritative parenting compared to female ones [35,38].

As shown above, the scarce research conducted into parenting styles during emerging adulthood raises more questions than answers. Another interesting issue that has yet to be clarified is which parenting style is more frequent during this period. In the classic study by Shucksmith, Hendry and Glendinning [40], the authors claim that during early adolescence the authoritative and authoritarian styles are more frequent, whereas the permissive and neglectful styles are more common later on, as control is gradually reduced, during the teenage years. However, the sample used in that study ranged only up to age 16, and no information is provided regarding what happens after that moment. Although some authors suspect that the permissive and neglectful styles may be more common during emerging adulthood, recent studies, such as the one by Alt [30] carried out with an Arabian sample, have found that undergraduate students perceived their parents as more authoritative than permissive or authoritarian.

The general aim of the present study is to analyze the relationship between parenting style, psychological wellbeing and psychological distress among emerging adults from Spain (SP) and Portugal (PT). As explained above, very few studies have analyzed parenting styles during emerging adulthood, and none has been done in any of these two countries. Yet, as Arnett himself points out [41], the way in which young people navigate towards adulthood and their experiences during this period vary greatly according to the macro cultural context in which they live and the social groups to which they belong. In accordance with this idea, the present study explores parenting styles in Spain and Portugal, two Southern European Countries in which the family context is particularly important during emerging adulthood.

There are a number of different reasons why the family context is so important in Portugal and Spain. Firstly, high levels of youth unemployment and unstable jobs make it very hard for young people to become financially independent. In fact, in both countries, over 80% of those under 29 years of age continue to live in the family home [42,43]. Secondly, poor and insufficient social polices force families to continue to financially maintain their older children [44]. And finally, the strong Catholic tradition in these countries strengths children’s ties to their families [44,45,46]. In such a context, it is hardly surprising that young people in the south of Europe attach greater importance to maintaining familial bonds than to becoming independent adults [47,48].

High levels of youth unemployment, scarcity of social support and strong and traditional family ties just described set up what Vogel [49] designated as a “family welfare regime”. This is a common feature of Portugal and Spain and is likely to have an impact on parenting styles and their effects on sons and daughters during emerging adulthood. Thus, we believe it is vital to analyze the role of parenting style among emerging adults in both of these countries to further understand how the family welfare regime may have an effect in emerging adults. However, it is worth noticing that both countries, despite sharing some aspects such as a family culture of emotional and financial dependence [46,50], nevertheless are different societies and have idiosyncratic cultural features [51]. Cross-cultural comparison may, then, offer interesting results concerning similarities and differences between the two contexts.

The present study has two principal aims. Firstly, it seeks to analyze the distribution of parenting styles among Spanish and Portuguese university students, in order to determine which are the most frequent during emerging adulthood; and secondly, it seeks to analyze the relationship between parenting style, psychological wellbeing and psychological distress among emerging adults in both countries. Possible sex-related differences will also be taken into account in all analyses.

## 2. Methods

### 2.1. Participants

The sample comprised a total of 1044 emerging adults aged between 18 and 30. The sample from Portugal comprised 491 undergraduate students from Porto University (213 women) with an average age of 20.29 (SD = 2.13) who formed part of the project Relações familiares em Portugal e ajustamento psicológico: investigação intercultural entre Espanha e Portugal. The sample from Spain comprised 552 undergraduate students from Seville University (282 women) with an average age of 20.20 (SD = 2.10) who formed part of the projects La transición a la adultez en España: Estudio sobre las claves del ajuste psicosocial y fundamentos para su intervención preventiva (EDU2013-45687-R) and Estudio longitudinal secuencial sobre la transición a la adultez en España (RTI2018-097405-B-I00). Samples were representative of the five major knowledge areas: arts and humanities, sciences, health sciences, social and legal sciences and engineering and architecture.

### 2.2. Procedure

After the purpose of the study was explained to faculty members from the two participating universities, and permission was requested to gather information from their students, approval to conduct the study was obtained. Once the appropriate consents had been obtained from the students, they completed a questionnaire package in the presence of a member of the research team. The questionnaires were administered in classrooms and took approximately 30 minutes to complete. The study was approved by the Comite de Ética de la Faculdade de Psicologia e de Ciencias da Educação da Universidade do Porto in Portugal (Refª 2017/10-2) and by the the Coordinating Committee for the Ethics of Biomedical Research in Andalusia (Spain) (ethical code).

### 2.3. Measures

*Demographic variables.* The demographic information questionnaire asked participants about their age, sex and field of study, and whether or not they lived in the family home.

*Parental warmth.* Participants completed the Spanish and Portuguese language versions (α = 0.85 in Portugal; α = 0.82 in Spain) of the six-item parental warmth subscale of the Perceptions of Parents Scales (POPS), College Student Version [52,53]. In this subscale, items were rated on a Likert-type scale ranging from 1 (“not at all true”) to 7 (“very true”). Example item: “My parent accepts me and likes me as I am”. High scores indicate high levels of perceived parental warmth.

*Parental behavioral control.* Participants completed the five-item version of Kerr and Stattin’s Control Subscale [54], adapted for emerging adults and translated into Spanish and Portuguese. The Cronbach’s alpha for this scale was α = 0.79 in Portugal and α = 0.77 in Spain. Items were rated on a Likert-type scale ranging from 1 (“completely disagree”) to 6 (“completely agree”). Example item: “My father/mother tries to control how I spend my money”. High scores indicate high levels of perceived behavioral control.

*Psychological Distress.* Participants completed the Spanish [55] and Portuguese [56] adaptations of the 21-item Depression Anxiety Stress Scale, reduced version (DASS-21), developed by Lovibond and Lovibond [57]. In this measure, participants rated each item on a Likert-type scale ranging from 1 (“does not apply to me at all”) to 4 (“applies to me very much, or most of the time”). The DASS-21 has three subscales: depression, anxiety and stress, which are grouped into a second order factor called general psychological distress (α = 0.92 in Portugal; α = 0.89 in Spain). Example item: “I felt sad and depressed”. High scores indicate high levels of psychological distress.

*Psychological wellbeing.* Participants completed the Spanish [58] and Portuguese [59] adaptations of the 29-item Psychological Wellbeing Scale (PWBS), reduced version, developed by Ryff, Lee, Essex and Schmutte [60]. Items were rated on a Likert-type scale ranging from 1 (“completely disagree”) to 6 (“completely agree”). The PWBS has six subscales grouped into a second-order factor called general psychological wellbeing [61]. The Cronbach’s alpha for this scale was α = 0.89 in Portugal and α = 0.87 in Spain. Example item: “When I look at the story of my life, I am pleased with how things have turned out”. High scores indicate high levels of psychological wellbeing.

### 2.4. Data Analysis

Statistical analyses were performed using the IBM Statistical Package for Social Sciences 24.0 software [62]. Parenting style categories were calculated separately for Spain and Portugal, with a different cut-off point being established for each country, since the means and standard deviations for parental warmth (M = 6.03, SD = 1.03 and M = 5.90, SD = 1.11 for Spain and Portugal, respectively) and behavioral control (M = 2.29, SD = 1.10 and M = 2.05, SD = 0.99 for Spain and Portugal, respectively) were statistically different (*t* = 2.05, *p =* 0.04, Cohen’s d = 0.12 and *t* = 3.67, *p* < 0.001, Cohen’s d = 0.23 for warmth and control, respectively). Parenting styles were defined using the method outlined in the classic study by Lamborn et al. [7], more recently used by Musitu and García [63] and García and Gracia [64]. Thus, the authoritative style is the one that scores in the top tercile in both dimensions (behavioral control and warmth); the neglectful style scores in the lower tercile in both dimensions; the authoritarian style scores in the upper tercile in behavioral control and the lower one in warmth; and finally, the permissive style scores in the upper tercile in warmth and in the lower tercile in behavioral control. In this procedure, mid-range values are not taken into account, meaning that the analyses were carried out with just 44% of the total sample. However, this procedure allows a “purer” way of obtaining parenting styles and, consequently, more accurate results.

The next step was to calculate the frequency of each parenting style by country, sex and experience leaving home, and to determine the Chi-squared values in order to identify any possible differences in parenting style in accordance with those variables. Finally, one-way analyses of variance (ANOVAs) were carried out, followed by a Bonferroni post hoc test to examine any possible differences between parenting style groups and emerging adults’ psychological wellbeing and psychological distress. Effect sizes (Cramer’s V, ω^2^ or Cohen’s d) were calculated and interpreted following the criteria described by Cohen [65] (*d* = 0.2 small, *d* = 0.5 medium and *d* = 0.8 large effect size).

## 3. Results

The descriptive analyses presented in the data analysis section revealed that emerging adults perceived high levels of parental warmth, with mean scores of 6 or close to 6 within a possible range of 1 to 7. In contrast, they perceived low levels of behavioral control, with mean scores of around 2 within a possible range of 1 to 6.

In relation to the first aim of the study, similar distributions of parenting styles were found in both countries analyzed (Figure 1). Thus, in both Spain (SP) and Portugal (PT), authoritarian parenting was the most frequent (15.2% SP and 14.2% PT), followed by the permissive style (13.2% SP and PT). Finally, in Spain, the frequency of the neglectful (7.2%) and authoritative (7.1%) styles was similar, while in Portugal, the neglectful style was the third most common (11.4%), and the least frequent was the authoritative style (6.5%). No significant differences were found in the distribution of parenting styles in accordance with country (χ^2^ (3) = 4.73, *p* = 0.193).

It is important to remember that, due to the procedure used to create the four parenting style categories, more than 50% of the sample was classified as missing values. It is also worth bearing in mind that control levels were low throughout the entire sample (with a mean of 2.29 in Spain and 2.05 in Portugal, on a scale of 0–6). Consequently, we should not think of authoritarian families as exerting a high level of control, but rather as exerting a low level of control that is nevertheless higher than the average in the sample.

Sex differences in the prevalence of parenting styles were found in Spain (χ^2^ (3) = 9.98, *p =* 0.02; Cramer’s V = 0.21) and Portugal (χ^2^ (3) = 13.80, *p =* 0.003; Cramer’s V = 0.25), with a moderate-large effect size. Among men, the permissive parenting style was less frequent (in SP and PT) and the neglectful one more frequent (in SP) than would be expected by chance (Figure 2).

Moreover, results showed differences according to leaving-home experience (χ^2^ (3) = 12.42, *p =* 0.006, Cramer’s V = 0.23, SP and χ^2^ (3) = 7.83, *p =* 0.05, Cramer’s V = 0.19, PT), with moderate effect size. In both samples, Portuguese and Spanish, there were more people than randomly expected living with their parents among the authoritarian group. Only in the Spanish sample, there were more emerging adults who had already left home than randomly expected among the permissive parenting group, a pattern that was also observed in the Portuguese sample among the neglectful parenting style group.

To determine how parenting styles are related to wellbeing and general distress, a series of one-way ANOVAs were conducted. To answer the study questions, the values were first calculated separately for Spain and Portugal, and then separately for men and women. The results revealed that, in both countries, parenting style is associated with both emerging adults’ wellbeing and distress. Emerging adults’ wellbeing (*F*(3,232) = 20.88, *p <* 0.001, ω^2^ = 0.20 SP; and *F*(3,218) = 33.23, *p <* 0.001, ω^2^ = 0.30 PT), with a large effect size, and their distress (*F*(3,231) = 8.11, *p* < 0.001, ω^2^ = 0.08 SP; and *F*(3,219) = 7.21, *p <* 0.001, ω^2^ = 0.09 PT), with a moderate effect size.

Bonferroni pair-wise comparisons were calculated in order to determine group differences between different parenting styles and wellbeing and distress. In relation to wellbeing (Figure 3), similar differences were found in both countries: neglectful versus permissive styles (mean difference = −0.44, *p* < 0.001, Cohen’s d = −0.81 SP; and mean difference = −0.62, *p* < 0.001, Cohen’s *d* = 1.07 PT), with a large effect size; neglectful versus authoritative styles (mean difference = −0.36, *p* = 0.04, Cohen’s d = −0.63 SP; and mean difference = 0.72, *p* < 0.001, Cohen’s d = 1.26 PT), with a moderate-large effect size; permissive versus authoritarian styles (mean difference = 0.67, *p* < 0.001, Cohen’s d = 1.17 SP; and mean difference = 85, *p* < 0.001, Cohen’s d = 1.42 PT), with a large effect size; and authoritarian versus authoritative (mean difference = 0.59, *p* < 0.001, Cohen’s d = −0.99 SP; and mean difference = 0.94, *p* < 0.001, Cohen’s d = 1.60 PT), with a large effect size. These findings indicate that the authoritative and permissive parenting styles are those most closely associated to a high level of wellbeing, with no differences being found between them; for their part, the neglectful and authoritarian styles are linked to lower wellbeing scores, again with no differences being found between them.

In relation to general distress (Figure 4), the Bonferroni test revealed higher distress levels in the authoritarian group than in the permissive one in both Portugal (mean difference = −16.24, *p* < 0.001, Cohen’s d = 0.80) and Spain (mean difference = 14.32, *p* < 0.001, Cohen’s d = 0.63), with large and moderate effect sizes, respectively. Furthermore, and only in Spain, in comparison with all the other groups, higher distress levels were found in the neglectful and authoritative groups. Neglectful group (mean difference = 11.69, *p* = 0.04, Cohen’s d = 0.51), with a moderate effect size, and the authoritative group (mean difference = −17.15, *p* < 0.001, Cohen’s d = 0.76), with a large effect size. Thus, the style most closely associated to distress among emerging adults was clearly the authoritarian one, both in Portugal and in Spain.

Given the similarities found between Spain and Portugal as regards the effect of parenting style on wellbeing and distress among emerging adults, the two samples were combined in order to analyze possible sex differences. The results revealed that, among the emerging adults of both countries, women from neglectful families scored higher for wellbeing than their male counterparts (*t*(94) = −2.47, *p* = 0.015, Cohen´s d = 0.51 ), with a medium effect size. No sex differences were found in the association between the other parenting styles and wellbeing: permissive (*t*(81.5) = 0.01, *p =* 0.99); authoritarian (*t*(151) = 0.31, *p =* 0.76); and authoritative (*t*(69) = 0.52, *p =* 0.61), indicating that the association between wellbeing and the authoritarian, authoritative and permissive styles was the same regardless of the sex of the child.

Similarly, the results failed to indicate any sex differences in the association between parenting styles and distress: neglectful (*t* (94) = −1.59, *p =* 0.116); permissive (*t*(136) = −0.55, *p =* 0.58); authoritarian (*t*(151) = −1.33, *p =* 0.186); and authoritative (*t*(69) = −0.34, *p =* 0.736) (Figure 5).

## 4. Discussion

The deep-rooted social and economic changes that have taken place in (particularly) Western societies over recent years have given rise to a new developmental stage called emerging adulthood [25,26]. As a result of this new reality, researchers need to take a long, hard look at the young people of the 21st century, in order to understand who they are and how they vary across cultures, since only in this way will we be able to help optimize their development.

This analysis must necessarily take into account the family context, for two main reasons. Firstly, it is important to understand what happens during this period in terms of parent–child relations. As children reach emerging adulthood, their relationship with their parents needs to change in order to adjust to the fact that although due to their socioeconomic situation they are still fairly dependent on their parents, they are nevertheless mature adults at many levels [28,29]. Secondly, it is important to take families into account because research has shown that this context continues to play a fundamental role in young peoples’ development, even during the third decade of their lives [66,67,68]. The present study aims to shed some light on family relations during this period among two samples of university students in Spain and Portugal by analyzing parenting styles and their implications for emerging adults’ psychological wellbeing and adjustment, with a special focus on sex differences.

One striking result found in our study is the high level of warmth and low level of control perceived by emerging adults belonging to the samples of both countries. As stated above, during emerging adulthood the family system needs to adjust to respond to the changing needs and demands of offspring who, while still dependent in many ways, are nevertheless no longer adolescents but rather adults [69,70,71]. Reducing the level of control they exert over their children is probably one of the main adjustments that should be made by parents during this period, and our results concur with those reported by other studies that also found low and medium levels of control during this stage [72,73]. The high levels of warmth perceived by the young people in our sample are also consistent with the findings of previous studies, which reported positive family relations during this period [74,75], with young people perceiving adequate levels of parental warmth [76] and intense parental involvement [77].

The most frequent parenting style among youth belonging to samples of both countries was the authoritarian one. However, this does not mean that the majority of Spanish and Portuguese emerging adults live in homes with high levels of control and low levels of warmth. As explained in the method section, the process by which the four parenting styles are established eliminates almost half of the sample, since it selects only those with scores located in the extreme terciles. Thus, when we talk about the authoritarian style, we are in fact talking about only 14% of the Portuguese sample and 15% of the Spanish one. Moreover, it should not be forgotten than although emerging adults report high levels of control and low levels of warmth, they only do so in relation to the mean. Nevertheless, the authoritarian style is the most frequent, being twice as common as the authoritative one, and it is particularly common among university students of both samples who still cohabite with their parents. This is hardly surprising, given the traditional conception of family that is dominant in both countries [45,46,78], which may make it difficult for parents to stop exerting control over their children [19,20]. Parents may also be more controlling than their children would like, especially if they still live under the same roof of their parents.

This high level of control may have a negative impact on family dynamics, giving rise to tenser relations and generating less affection and warmth between parents and children; alternatively, it may be that although parents really do offer support and affection, their older children are simply unable to perceive it. Whatever the case, this is the first study to analyze parenting styles in Spain and Portugal during emerging adulthood, and it would be useful to try and replicate the results among the general population (not just university students), and to include also parents’ perspectives.

In our opinion, the most interesting finding here is the significant association observed between parenting style and adjustment among emerging adults of both samples, since parenting style is clearly linked to both psychological wellbeing and psychological distress. In both, Portugal and Spain, the authoritative and permissive styles are those most closely associated with high levels of wellbeing, while children who perceive their parents as neglectful and authoritarian scored lowest in this variable. These findings coincide with those reported by other studies carried out with children and adolescents in Latin American and Southern European countries, where the permissive style was found to be just as positive for wellbeing as the authoritative one [5,7,64,79,80,81,82]. These results suggest that parental control and supervision may not actually be welcomed, and highlight the positive effect of more symmetrical relationships based on warmth and affection during emerging adulthood.

In fact, other studies conducted with Spanish adolescents have concluded that behavioral control is fundamental to good development [83,84]. However, the present study, carried out with emerging adults, clearly indicates that the permissive style is as closely linked to wellbeing as the authoritative one, thereby underscoring the importance of parental warmth for ensuring adequate development during this life stage. Our results regarding the association between parenting style and psychological distress are similar in nature. In both Spain and Portugal, the style most closely related to distress among emerging adults of our samples is the authoritarian one, which is characterized by a high degree of control and a low level of warmth.

It is likely that the negative effect of control, which when combined with low levels of warmth is linked to psychological distress, is due to the age of the young adults in our sample. As mentioned in the introduction, the behavioral control exercised by parents tends to diminish as adolescence progresses [40,84], with children being granted more autonomy as they approach adulthood. It is important to underline that all of our participants formally reached the age of majority at the time of the study. Thus, these results may suggest that their perceptions of parents, who still sought to control them, may have been linked to a lack of trust by the parents in their children’s capabilities, and an unwillingness to grant the children autonomy. This control, coupled with a low level of perceived warmth, could have had a detrimental effect on the children’s psychological wellbeing.

One possible explanation for the positive role played by the permissive style can be found in García and Gracia [85]. According to these authors, in horizontal collectivist cultures such as Portugal and Spain, relationships are expected to be more egalitarian and symmetrical than in vertical collectivist (Asian or Arabian) or individualistic cultures (North American). Thus, in Southern European countries, the use of power, imposition and control would be perceived by children as coercive rather than as a necessary component of care and responsibility, and as such, are not linked to any indicator of positive adjustment. If, as these authors argue, this is already the case during adolescence, then the feeling would only be more intense during emerging adulthood, when children have become adults and demand greater levels of autonomy and independence.

As regards sex differences, our results indicate a fairly similar situation among men and women, both in regards to their perception of parenting styles and in relation to the association between those styles and psychological wellbeing and distress. In reference to perceived parenting styles, although the distribution was similar between men and women, men in our sample did report less permissive and more neglectful styles than would be expected by chance. Nevertheless, the authoritative and authoritarian styles were distributed similarly among men and women, with the percentages for both styles in both sexes being similar in Portugal and Spain, thereby indicating, once again, important likenesses between the two contexts studied.

Many other similarities were also found in relation to the link between parenting style and psychological wellbeing and distress among men and women, with the authoritarian style being the most harmful for both sexes. Moreover, the permissive and authoritative styles were most closely related to wellbeing in both male and female emerging adults of our sample. These similarities between men and women in terms of how parenting styles affect their psychological wellbeing and distress suggest that the effect of parenting style during this period is so clear that it goes beyond the gender of the child. In fact, at least in Spain and Portugal, the most beneficial and most harmful styles are the same for both male and female emerging adults.

Turning to the differences between Portugal and Spain, our results indicate that the situation is fairly similar in both contexts. Parenting styles were distributed in the same way, with the authoritarian and permissive styles being more prevalent among youth of our study, and the authoritative and neglectful ones less so. Furthermore, as mentioned earlier, the relationship between parenting style and the adjustment of emerging adults was similar also. These similarities can probably be explained, at least partially, by what Vogel [49] calls “family welfare regimes”, a label that refers to Southern European countries characterized by high youth unemployment rates and low levels of social investment, strong traditional family bonds and high levels of poverty and social inequality. Given that within these parameters (which are shared by both Spain and Portugal) the family acts as the principal provider of resources and security, even during emerging adulthood [78,86], it is hardly surprising that the two contexts are similar in regards to the effect of parenting style on older children’s development. Obviously, Portugal and Spain are different countries, with social and economic differences. However, in terms of family relationships and their effect on emerging adults’ wellbeing, the similarities outweigh the differences.

Despite these interesting results, our study, like all research, has certain limitations. Firstly, the sample was relatively small, and was made up exclusively of university students, 90% of whom were aged under 24. Consequently, we must be cautious when generalizing our findings to older emerging adults and those who are not studying at university. Future studies should strive to include a larger sample that is representative also of the non-university population, and broaden the age range to include the entire emerging adult age range (up to 29 years of age). The second limitation is that, since we did not distinguish between maternal and paternal parenting styles, we could determine whether the effect of the style differs in accordance with the sex of the parent exercising it (the father or the mother). Linked to the above, a third limitation is that we did not include parents’ perspectives in our analyses. Taking the viewpoint of the other stakeholders involved in the relationship into account would have enabled us to triangulate the information and gain a more complete and comprehensive view of the family relations present in each case. Future studies may therefore include parents’ views in the analyses and explore children’s perspectives of their mothers’ and fathers’ parenting styles separately, in order to come to a more in-depth understanding of family dynamics during this period. The final limitation is the fact that the study was cross-sectional in nature, preventing us from establishing the direction of causality in the relationships observed. Thus, for example, although it may be that the authoritarian style does indeed generate more psychological distress, it may also be that children with higher levels of emotional distress perceive their family relationships more negatively and their parents as being more authoritarian. Alternatively, it may even be that the symptoms of psychological distress provoke parental dynamics characterized by more emotional distance and stricter control [87]. Future studies should consider using longitudinal designs, which would enable causal relations to be determined more accurately.

These limitations notwithstanding, we believe the present study sheds some light on a question that has received very little attention to date in Southern European countries such as Portugal and Spain, namely the effect of parenting style on children’s development during emerging adulthood. If we had to highlight one finding in particular, it would be, without a doubt, the positive effect of a family climate characterized by support and affection during this period, and the clearly negative effect of control. Thus, the most beneficial styles were the authoritative and permissive ones, and the most harmful one the authoritarian. This finding has clear practical implications in terms of parent training. As stated at various points in this paper, emerging adulthood is a new developmental stage in which young people, despite being adults, continue to live in semi-adolescent conditions in the family home. As such, parents have no references regarding the best ways to treat their children during this new stage [72]. It is therefore the responsibility of researchers to provide the data required to develop evidence-based intervention programs designed to help mothers and fathers exercise their parental role in the best possible way during this period, since we know that what parents do and do not do can have a huge impact on their children’s development.

Our findings clearly indicate that, in Spain and Portugal, as well as in other countries, parents should continue to support their children in a climate of warmth and affection during emerging adulthood, while at the same time reducing their levels of control. At least, in what concerns to control as it was exercised in previous years, since it should take the form of guidance and advice rather than behavioral supervision or rule setting during the third decade of life. In this sense, recent studies have analyzed the effect of “helicopter parenting” on the wellbeing of adults and emerging adults [88,89]. It is important to analyze in more detail the role of control/advice/guidance during this developmental stage, in order to determine whether or not it is necessary, and, if so, what the best way of providing it is.

Finally, future research should continue to explore the keys to promoting positive parenting during emerging adulthood, broadening samples to include non-university students and taking into account the view of parents. Studies should also adopt a longitudinal perspective covering the entire third decade of life. Only in this way will we be able to support positive parenting during emerging adulthood, thereby contributing to the psychological wellbeing of young people and the improvement of their family relations.

## Figures and Tables

**Figure 1 ijerph-16-02757-f001:**
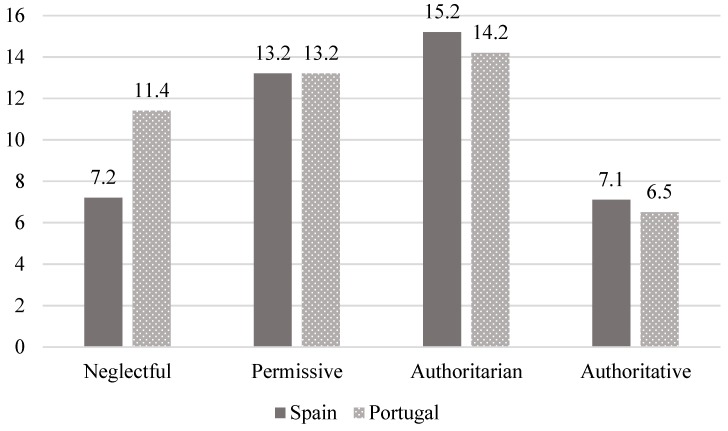
Parenting style distribution by country.

**Figure 2 ijerph-16-02757-f002:**
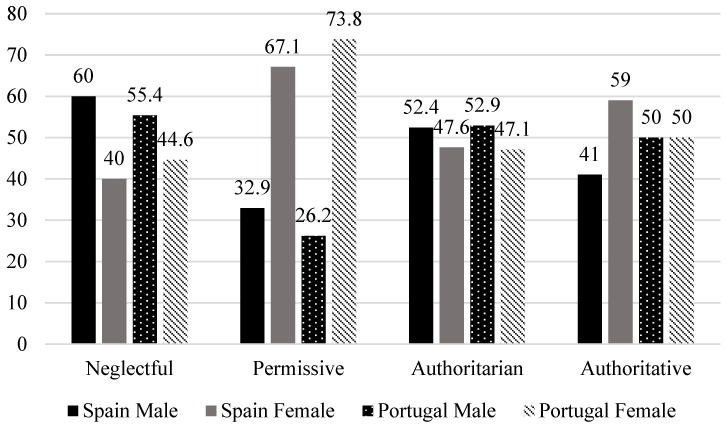
Sex distribution among each parenting style in Spain (SP) and Portugal (PT).

**Figure 3 ijerph-16-02757-f003:**
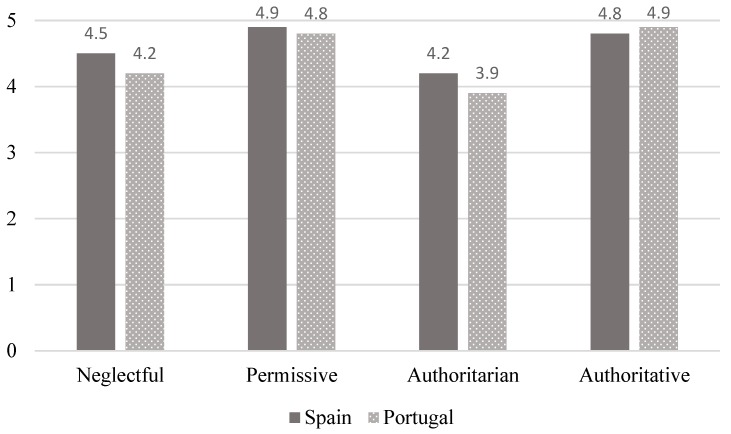
Differences in wellbeing according to parenting styles in Spain and Portugal.

**Figure 4 ijerph-16-02757-f004:**
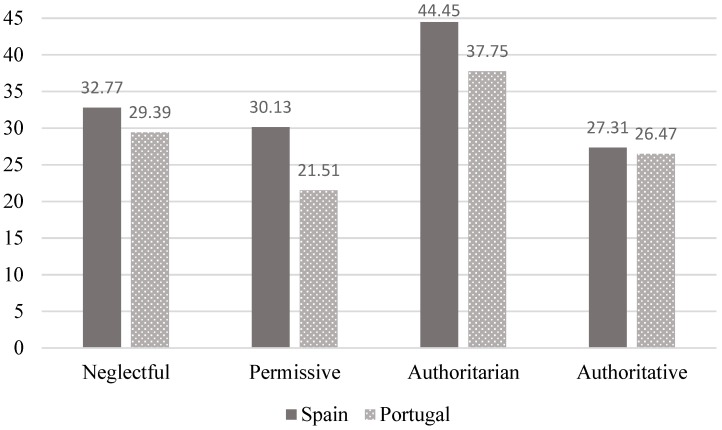
Differences in psychological distress according to parenting styles in Spain and Portugal.

**Figure 5 ijerph-16-02757-f005:**
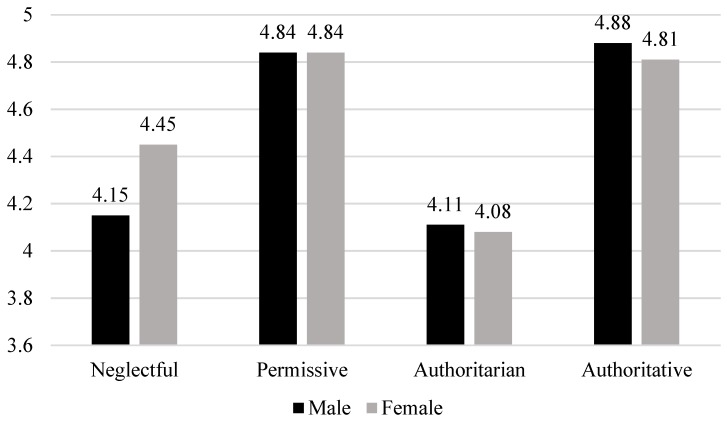
Relations between parenting styles and wellbeing in males and females.

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
