# Peer review of "Perceived Parenting Styles and Adjustment during Emerging Adulthood: A Cross-National Perspective"

_ijerph, 2019, doi:10.3390/ijerph16152757_

Round 1

Reviewer 1 Report

This was a very straightforward paper. Here are my suggestions:

- Add "Perceived" in the title as this study looked at perceived parenting style.

- Use "association" and avoid terms such as "relationship" when you work with categorical data.

- On p.7 line 237 you said "...more than 50% of the sample was classed as missing values". This is a huge loss of data and you need to take that into account. You need to add one more section under your methods part and explain how you dealt with this. I string suggest using imputation methods to take care of the sample. If it didn't work and you ended up working with the smaller sample then definitely use bias-corrected bootstrapped confidence interval for all of the hypothesis testings and report them.

- Report effect size fro all significant ANOVAs. I recommend Omega squared.

- Make sure everything is in English. Figure 2's caption is in Spanish.

- One thing that really concerns me is the fact that you used a (relatively small) sample from two universities and you are generalizing the results to the two countries. I couldn't find any rational and support for sample representativeness. The whole "discussion" section talks about "both countries" but in fact, we're talking about participants in two universities in two countries. Since this study was not based on census or a country-wide representative sample, I strongly suggest using a milder language and limit your discussion/conclusion to the sample or their associated population.

Author Response

We are very thankful for all the comments and suggestions made by the reviewers concerning our manuscript entitled Perceived Parenting Styles and Adjustment during Emerging Adulthood: A Cross-National perspective. All of the comments and suggestions have undoubtedly contributed to the improvement of its quality and adjustment to the standards of IJERPH.

We will now proceed with the exposition of all the changes that have been made in the manuscript in order to meet the suggestions of the reviewers.

Reviewer 1

This was a very straightforward paper. Here are my suggestions:

- Add "Perceived" in the title as this study looked at perceived parenting style.

Response: Done

- Use "association" and avoid terms such as "relationship" when you work with categorical data.

Response: Done

- On p.7 line 237 you said "...more than 50% of the sample was classed as missing values". This is a huge loss of data and you need to take that into account. You need to add one more section under your methods part and explain how you dealt with this. I string suggest using imputation methods to take care of the sample. If it didn't work and you ended up working with the smaller sample then definitely use bias-corrected bootstrapped confidence interval for all of the hypothesis testings and report them.

Response: Concerning data imputation, it is important to underline that there are no missing data. The scores of those participants that are lost are due to the procedure used to calculate the parenting styles, based on the terciles, as it is explained in the text.

- “Parenting styles were defined using the method outlined in the classic study by Lamborn et al. (1991). Thus, the authoritative style is the one that scores in the top tercile in both dimensions (behavioral control and warmth); the neglectful style scores in the lower tercile in both dimensions; the authoritarian style scores in the upper tercile in behavioral control and the lower one in warmth; and finally, the permissive style scores in the upper tercile in warmth and in the lower tercile in behavioral control. In this procedure, mid-range values are not taken into account, meaning that the analyses were carried out with just 44% of the total sample.” (p.5, l 216-233)

Response: Furthermore, we have also made the analysis using bootstraping in SPSS software. The results remain the same. Yet, the results’report and understanding, by the reader, may be much more challenging. Considering this, we think that it would be more parsimonious to leave the text as it was in its original version.

- Report effect size fro all significant ANOVAs. I recommend Omega squared.

Response: Done

- Make sure everything is in English. Figure 2's caption is in Spanish.

Response: Done

- One thing that really concerns me is the fact that you used a (relatively small) sample from two universities and you are generalizing the results to the two countries. I couldn't find any rational and support for sample representativeness. The whole "discussion" section talks about "both countries" but in fact, we're talking about participants in two universities in two countries. Since this study was not based on census or a country-wide representative sample, I strongly suggest using a milder language and limit your discussion/conclusion to the sample or their associated population.

Response: Discussion was modified in order to make it more prudent regarding the generalization of the results. 

Reviewer 2 Report

“Parenting Styles and Adjustment during Emerging 3 Adulthood: A Cross-National perspective” examines how child-reported warmth and control interact to predict psychological functioning among college students in Spain and Portugal. The manuscript is well-written and examines an interesting research question in a novel population. The introduction does a great job in introducing parenting styles and emerging adulthood, and then clearly lays out why the two might be correlated in southern European young adults. The authors are should be commended for their use of a large, cross-national sample.

There are some limitations that can be remedied before this manuscript is published. Two of these changes are in analytic approach, and one is a small theoretical matter.

Minor Revisions:  

(1)

The authors’ thinking is clearly informed by the “family welfare regime” and how it impacts emerging adulthood in Spain and Portugal, which I think is strength of the paper. I would like a little more explicit consideration of  how the family welfare regime might impact parenting styles, especially in the introduction.

(2)

The authors collect data on whether the participants live in the family home, but do not report whether these findings descriptively or inferentially. I think this variable might be more important to the relationship between parenting styles and functioning than the gender moderation question that the authors explored. The ability to expert parental control is likely much stronger when a child lives at home than when a child lives on or near their college campus.  I think this paper would be stronger if it reported results relating to whether or not children live in the family home.

(3)

I understand that the authors are following the conventions of the parenting styles literature by using a tercile split procedure. However, I think that this procedure is deeply flawed. Simply disregarding the responses of 56% of respondents is both a waste of the researchers time collecting data and ensures a much less stable parameter estimate of the observed effect.

 I think these analyses could make a much stronger claim if they were carried out as a simple two-way interaction (with warmth and control predicting psychological functioning). Based on the results reported here, it seems as if the authors might find a warmth effect with null findings for control and the interaction. However, this is nearly impossible to discern, given that the authors discard the majority of their data.

            If the authors do keep the tercile split procedure, there should be much stronger theoretical rational given (e.g. parenting styles only matter on the edges of the distribution, that 50th percentile warmth is functionally indistinguishable from 66th percentile warmth) rather than simply following the lead of a study conducted 28 years ago. 

Line edits:

Page 11, Line 421: I do not think you need the “traditonial” parenthetical, which reifies a hetero-normative default for parenting relationships

Author Response

Reviewer 2

Comments and Suggestions for Authors

“Parenting Styles and Adjustment during Emerging 3 Adulthood: A Cross-National perspective” examines how child-reported warmth and control interact to predict psychological functioning among college students in Spain and Portugal. The manuscript is well-written and examines an interesting research question in a novel population. The introduction does a great job in introducing parenting styles and emerging adulthood, and then clearly lays out why the two might be correlated in southern European young adults. The authors are should be commended for their use of a large, cross-national sample.

There are some limitations that can be remedied before this manuscript is published. Two of these changes are in analytic approach, and one is a small theoretical matter.

 Minor Revisions:  

(1)    The authors’ thinking is clearly informed by the “family welfare regime” and how it impacts emerging adulthood in Spain and Portugal, which I think is strength of the paper. I would like a little more explicit consideration of how the family welfare regime might impact parenting styles, especially in the introduction.

Response: The introduction was briefly expanded in order to meet the reviewer’s suggestion. It was not possible to develop more broadly because of the limits of words for the paper (it already has 9293 words), and also because we believe that Vogel’s (2002) contribution for the characterization of “family welfare regime” is further described in the discussion.

(2)    The authors collect data on whether the participants live in the family home, but do not report whether these findings descriptively or inferentially. I think this variable might be more important to the relationship between parenting styles and functioning than the gender moderation question that the authors explored. The ability to expert parental control is likely much stronger when a child lives at home than when a child lives on or near their college campus.  I think this paper would be stronger if it reported results relating to whether or not children live in the family home.

Response: This analysis was included both in the results and in the discussion sections. It has been done briefly in order to avoid that the manuscript becomes too extensive and also because the impact of living or not in the family home, in spite of its importance, was not one of the aims of this manuscript. The effect of cohabitation with parents has been developed in other previous publication of our research team (c.f., García-Mendoza, M.C., Parra, A., Sánchez-Queija, I. (2017). Relaciones familiares y ajuste psicológico en adultos emergentes universitarios españoles. Behavioral Psychology/Psicología Conductual, 25(2), 405-417. 

(3)    I understand that the authors are following the conventions of the parenting styles literature by using a tercile split procedure. However, I think that this procedure is deeply flawed. Simply disregarding the responses of 56% of respondents is both a waste of the researchers time collecting data and ensures a much less stable parameter estimate of the observed effect.

 I think these analyses could make a much stronger claim if they were carried out as a simple two-way interaction (with warmth and control predicting psychological functioning). Based on the results reported here, it seems as if the authors might find a warmth effect with null findings for control and the interaction. However, this is nearly impossible to discern, given that the authors discard the majority of their data.

If the authors do keep the tercile split procedure, there should be much stronger theoretical rational given (e.g. parenting styles only matter on the edges of the distribution, that 50thpercentile warmth is functionally indistinguishable from 66th percentile warmth) rather than simply following the lead of a study conducted 28 years ago. 

Response: In spite of our overall coincidence with the reviewers’ suggestions and perspectives, we do not totally agree with this specific one. The special issue for which we have been invited of the International Journal of Environmental Research and Public Health is entitled Parental Socialization Styles in 21st Century and Adolescent Health and Well-being. Thus, the framework according to which the paper should be written is precisely the conceptualization of parenting styles. This conceptualization requires the creation of 4 parenting styles based on the affect and control dimensions. So, according to our point of view, it was not enough, as the reviewer suggests, to use only the affect and control dimensions and to assess the interaction effect between them.

Surely, to use tercile’s scores to obtain parenting styles will eliminate part of our sample, but it also guarantees that the styles are more “pure”. According to our point of view, to consider, for instance, one style as neglectful because it is a few tenths below the median on affect and control would be a risky and less accurate approach, comparing with our decision of only considering this style when the scores are clearly more extreme. Accordingly, we have preferred to be conservative and to create parenting styles eliminating the participants who had scores around the median. We added this explanation in the method section, as suggested by the reviewer.

Response: Finally, creating parenting styles based on terciles is not solely a classical approach adopted by authors such as Lamborn, Mounts, & Steinberg (1991) or Steinberg, Lambron, Darling, Mounts, & Dornbush (1994); in fact, more recent works of reputed researchers also used the same procedure, such as Musitu & García (2004) or García & Gracia (2009).

Line edits:

Page 11, Line 421: I do not think you need the “traditonial” parenthetical, which reifies a hetero-normative default for parenting relationships

Response: Done